# Field Emission Cathodes to Form an Electron Beam Prepared from Carbon Nanotube Suspensions

**DOI:** 10.3390/mi11030260

**Published:** 2020-02-29

**Authors:** Karolina Urszula Laszczyk

**Affiliations:** Wroclaw University of Science and Technology, Faculty of Microsystem Electronics and Photonics, 53-633 Wroclaw, Poland; karolina.laszczyk@pwr.edu.pl

**Keywords:** field emission, electron beam, carbon nanotubes

## Abstract

In the first decade of our century, carbon nanotubes (CNTs) became a wonderful emitting material for field-emission (FE) of electrons. The carbon nanotube field-emission (CNT-FE) cathodes showed the possibility of low threshold voltage, therefore low power operation, together with a long lifetime, high brightness, and coherent beams of electrons. Thanks to this, CNT-FE cathodes have come ahead of increasing demand for novel self-sustaining and miniaturized devices performing as X-ray tubes, X-ray spectrometers, and electron microscopes, which possess low weight and might work without the need of the specialized equipped room, e.g., in a harsh environment and inaccessible-so-far areas. In this review, the author discusses the current state of CNT-FE cathode research using CNT suspensions. Included in this review are the basics of cathode operation, an evaluation, and fabrication techniques. The cathodes are compared based on performance and correlated issues. The author includes the advancement in field-emission enhancement by postprocess treatments, incorporation of fillers, and the use of film coatings with lower work functions than that of CNTs. Each approach is discussed in the context of the CNT-FE cathode operating factors. Finally, we discuss the issues and perspectives of the CNT-FE cathode research and development.

## 1. Introduction

A focused beam of electrons is used in a wide spectrum of applications, including atomic-resolution imaging, chemical and crystallography analyses, cancer therapies, nanotechnology, and entertainment. It is implemented in electron microscopes, X-ray spectrometry, X-ray sources, and flat panel displays. Recently, there has been an increasing demand for miniaturized FE electron sources, especially for mobile devices [1,2,3,4,5] in order to extend their ability to work, e.g., in harsh and inaccessible environments. This progress might soon be expanded into novel hybrid devices, which might combine the miniaturized devices for energy harvesting, energy storage, and field emission [6]. The use of a miniaturized FE electron source could reduce the overall size of the final device. This is because FE electron sources, in contrast to thermionic electron sources, do not need a cooling unit or additional space to manage to heat to about 1000 °C.

The successful implementation of miniaturized electron sources depends on a smooth know-how transfer from a laboratory to a factory. This affects the price of the final product and the cost of the material and technology. On the other hand, commercial devices operating with the electron beam (~1950 for TEM, SEM, X-ray tubes, and many others) in most cases use tungsten as the primary building material for the electron sources, particularly for cathodes. It is extracted from commodities such as scheelite and wolframite and is harder than steel, more resistant to fracturing than diamond, and it withstands high temperatures (melting point: about 3400 °C). In a natural deposit, it is provided by China that covers about 80% of the total demand, while the remains are recycled (https://minerals.usgs.gov/minerals/pubs/commodity/tungsten/myb1-2013-tungs.pdf, accessed on 23rd January 2019). It is widely accessible for about 20 USD per unit, and it has been successfully applied in electron microscopes. In these circumstances, the lack of diversity among candidate materials limits the development of novel devices. Therefore, researchers have directed their interest toward novel and recently discovered materials with new outstanding properties, such as nanowires made of silicon carbide (SiC) [7], copper sulfide (Cu_2_S) [8,9], nanocarbons (carbon nanotubes (CNTs), graphene, etc.) [10,11].

Nanocarbons, in the form of highly viscous suspensions, are already present as commercial products and are being used in emerging applications that evolve into mobile and wearable electronics [12,13]. Viscous suspensions as fabrication material have opened new opportunities in terms of large-area and low-cost processability, especially in the case of complex systems. For example, industrial research has indicated that slurry screen printing is a potential technology for creating a large FE area. Therefore, after more than forty years, the FE array concept has been brought to light again [14,15], highlighted by its application for X-ray tubes [16,17] to achieve higher current densities and to avoid screening effect, which is present in case of densely grown CNT forest.

In this review, the author would like to take a closer look at FE cathodes for the electron sources based on viscous suspensions made of CNTs and review their design, technology, and performance, together with their integration into specialized instruments. The author believes the data included in this review will be useful for technologists and researchers from an interdisciplinary field who would like to widen their interest in the application of CNT suspensions to form a FE cathode, especially in terms of their miniaturization, multiplication, and arrangement, which might be extended from screen printing and contact techniques toward developing 3D printing. The review content includes basics of the FE cathodes (p. 2), with the evaluation of the cathodes (p. 3) and performance factors (p. 4), CNT as an electron-emitting material (p. 4), CNT suspension for FE cathodes (p. 5), including screen-printed CNT-FE cathodes (p. 9) and CNT-FE on the tip of a rod/wire (p. 12), methods to enhance field emission, including postprocess treatment (p. 13) and addition of fillers and coatings (p. 17), and finally, a summary (p. 19), issues, and perspectives (p. 22).

## 2. Basics of the FE Cathodes

In order to initiate an electron beam, an electron source is needed. In the simplest setup, an electron source is built of an opposing cathode and anode separated by a vacuum gap. Various electron sources can be used to emit electrons. In the literature, we can find electron emission sources based on thermal energy [18], field emission (FE) [19], Schottky emission [20], photoemission, and secondary emission [21]. Murphy and Good [19] identified that either one of the primary conditions (temperature or electric field strength) governs electron emission or that an intermediate region exists, where the temperature and electric field both contribute to the electron emission. In the case of thermal emission, the high temperature prevails. Typically, it is required for a cathode to be heated about 103 K [19,22]. The emission takes place over the barrier, and the emission current varies with temperature. In the case of cold field emission, high field strength dominates over temperature. Emitted electrons have energies below the Fermi level. The emission current varies with the electric field strength that determines the barrier shape. In field emission, electrons are emitted in the presence of a high electric field over 108 V/cm, at high or ultra-high vacuum (UHV; ~10^−5^–10^−10^ Torr), and through quantum tunneling at room temperature [23]. In UHV, the electrons do not collide, e.g., with the residual particles, and can travel far faster than in semiconductors, without dissipation of energy. The FE cathode, also known as a cold cathode, is used to emit electrons with high energy from keV to MeV from a solid surface. However, in Schottky’s sources, there is an ambiguous condition, i.e., the electric field is below 108 V/cm and acts together with thermal enhancement.

For an appropriate and thorough theoretical discussion, the author would like to refer to the tutorial papers on the electron sources by Jensen [21] and Forbes [24]. For field emission, usually, the Fowler–Nordheim (F–N) equation is commonly applied [19,20]. Albeit, there is still no decisive experimental evidence for the theoretical calculations [25]; the readers should consider works of Murphy and Good, or Forbes, and others who are not mentioned here, who reported the progress in correction of the F–N equation [19,25,26,27].

### 2.1. Evaluation

For the basic evaluation, the FE electron sources are measured in a diode configuration, i.e., cathode–anode, with a gap between them and with vacuum pressure below 10^−7^ Torr in a device or chamber for stable electron emission. Such a configuration is faster and more cost-efficient compared to the triode configuration, which is formed by a cathode, an extraction electrode/gate, and an anode. The triode and diode configurations both can be regarded as an electron gun [28]. The latter is used for more sophisticated assessment, e.g., if we need to steer the emission current or focus the electron beam. The measured sample is placed in the UHV chamber and heated to ~200 °C, in order to attain the required pressure and to ensure the presence of residual gases (coming from the cathode materials, e.g., not completely removed non-water solvent), and water is excluded or at least highly reduced. In such a condition, stable emission is expected, though it is not always the rule [29,30,31]. To visualize the electron emission, as an anode, the phosphor film is used deposited on a semi-transparent or transparent substrate. As a separator, it is made either a space gap, or a dielectric frame is used made of Teflon, or glass, and then placed between the cathode and the anode.

Usually, the principal characteristic is the current–voltage (I–V) graph, with the relation of the emitted current and applied voltage difference or electric field expressed in Vμm^−1^.

For an array of the field emitters, in most of the papers, including some recently reported [32,33], it is related to the Fowler–Nordheim law and it has a simplified form (considered as too simplified and inadequate—for more details please refer to [25]):*J* ≈ A∙E^2^/ϕ exp(−B∙ϕ^3/2^/E)(1)
where *J* is the current density, and E is applied local electric field at the cathode surface. E is related to the macroscopic electric field with the field enhancement factor, β; ϕ is the work function of the material—an intrinsic material property defined by the energy difference between the Fermi level and the vacuum level. A and B are constants, where A = 1.54 × 10^−6^ AV^−2^ eV and B = 6.83 × 10^7^ cm^−1^ V eV^−3/2^ [32].

Furthermore, to determine the field-emission properties, the I–V relation is often translated to the relation of the current density and the applied electric field (JE^2^ vs. 1/E), so-called “F–N coordinates” [26].

The resulted F–N plot is then approximately a linear curve, which indicates only that the emission process is probably F–N tunneling [25] The field enhancement factor, β, can be estimated from the F–N plot, using the following equation [34,35]:β = B∙ϕ^3/2^∙s^−1^(2)
where s is the slope of the F–N plot. The factor β depends on the emitting material geometry, the material crystallography [36,37], and the distance between the electrodes. In case of CNT film emitters, there is an additional difference that arises due to differences in morphology, chemical state, and variations in the experimental setup [38]. As the Fowler–Nordheim theory was derived for a flat surface, it has been proposed that a correction may be needed for surfaces when applied to a single carbon nanotube or for an emitting material shape with an extremely large curvature [39,40,41].

The author decided to compare only I–V and lifetime characteristics of various CNT-suspension-based FE sources. The reasons are as follows: (1) The review focuses on a technological aspect of the FE sources made of CNT-based suspensions, and (2) to avoid the misjudgment, considering the wide spectrum of the F–N equations and recent progress, as it has been mentioned at the begin of this section. Especially, as in recent reported scientific discussion [25], it was proposed to use the modified F–N equations named Murphy–Good equations and the Schottky–Nordheim barrier that represent better physics to explain and interpret the FE characteristics. The *Seppen–Katamuki* (*SK*) analysis should also be recalled here, as it enables us to obtain the exact work function of the emitter, as well as to extract geometrical parameters of the field emitter [42]; for example, *SK* analysis has been used to evaluate the changes in work function at elevated temperature [43] or to derivate the length of carbon nanotubes in the field-emission arrays [44].

In addition to the F–N plot, the emission current stability is evaluated for its lifetime stability, often referred to as a lifetime or aging test. The aging test is considered due to a few factors. One of them is residual ionized gases present in the cathode, surrounding and coming into physical and chemical reaction with the emitting material. Another one is degradation due to resistive heating, which takes place during emission. Resistive heating promotes the thermal decomposition of the emitting material due to high emission currents. It leads to the thermal instability of the cathode [45,46]. It has an influence on the electrical and thermal conductivity of the emitting material. On the other hand, it promotes emission. Hence, there might be a transition from field emission to thermal emission.

### 2.2. Performance Factors

To describe field emission quantitatively, few parameters are used, such as turn-on field, TOF, threshold field, Eth, or threshold voltage, Vth, with the corresponding total current or current density [45]. These parameters can be read from I–V plots. TOF describes the required applied electric field to switch on to achieve a target emission current density; usually, it is 10 μA/cm^2^. A low TOF means an emitter is characterized by low applied voltages to initiate emission, low power consumption, and a long lifetime [47]. In the literature, it is also known as a turn-on electric field [12]. It should be distinguished from the “Time of Flight” (ToF) measurement used to describe the time the object (particle, wavelength, etc.) needs to travel through a defined distance in a medium. It should be separated from the threshold voltage or threshold field, which represents the tip at which measured currents exceed 0.1 pA [39].

In the literature, authors usually assign the indicated current density to the electric field or voltage applied. Therefore, this electric field and voltage were here named as the subsequent threshold electric field, Eth, or the subsequent threshold voltage, Vth, following the definition from [48].

## 3. CNTs as Electron-Emitting Material

To achieve the efficient FE cathode, there are two factors to be considered: the material and the cathode shape. The material should possess a low work function, to enhance electron emission. For the cathode shape, a high aspect ratio structure is a common choice, because this intensifies the electric field. In addition, from a practical standpoint, the cathode material must be compatible with current technology so that it can be shaped and arranged with the other components into an electron gun at a relatively low cost.

As mentioned in the introduction, researchers are interested in novel nanomaterials that possess outstanding properties. One such nanomaterial is the quasi-one-dimensional CNT discovered by Iijima et al. in 1991 [49]. There are numerous advantages of CNTs: (1) a high aspect ratio with a small radius curvature of the tip, which is useful to generate a high-intensity electric field to kick out the electrons; (2) high thermal end electrical conductivity; (3) mechanical strength; and (4) thermal and chemical stability [50,51]. The work function for CNT equals to 5.1 ± 0.1 eV [34]. Therefore, shortly after CNTs were discovered, they were considered to be useful as the stable FE cathodes with promising long lifetimes [52,53] that can offer high brightness, outperforming the other sources by a factor of ten [54].

CNTs were mentioned as a potential material for FE displays [55], providing a current density of a few mA/cm^2^ under ultra-high vacuum (~10^−8^ mba or 10^−11^ Torr) [31,56], which is close to the required level of 10–100 mA/cm^2^ to ensure bright electroluminescence [45]. The mechanism of electron emission from CNTs, as well as the accompanying phenomena, is described widely in [23,57,58,59,60,61] and in many other studies. With the progress in understanding further CNT-based field emission, the developments of cathodes have focused on growth techniques of CNTs in order to (1) obtain a designable device configuration on large-area substrates, (2) obtain uniform emission, (3) overcome the field screening effect [62], and, finally, (4) overcome obstacles in reducing stability of CNT-based field emission in long-term testing [63].

Work on CNT-FE cathodes started to be conducted at an industrial site by Sony, in cooperation with Candescent Technologies Corp. [64], Samsung [65,66] (Figure 1a,b), and Philips [67] (Figure 1c). The research stopped because of the emergence of the OLED; the latest news about commercial field-emission displays (FEDs) comes from 2009 [68] and 2010 [69]. Despite the stoppage, CNTs and their composites continue to be researched in terms of cold emission, which is unveiling new phenomena and mechanism models [70,71,72,73] from doped CNTs [74], CNT fibers [75,76] and rods [77], or triangular spatial film [78].

## 4. CNT Suspension for FE Cathodes

There are various methods for synthesizing CNTs for FE electron source cathodes, e.g., arc discharge, chemical vapor deposition (CVD), and laser vaporization [79]. The first FE cathodes were isolated multi-walled CNTs grown by a plasma arc discharging directly onto a substrate, then attached to a graphite fiber electrode [80] (Figure 2a), and followed by accurately aligned arrays of CNT forests [52,57] (Figure 2b,c). FE cathodes can be formed as well by the deposition of synthesized CNTs in electrophoresis (Figure 2d,e) [81,82]. However, each of these methods are difficult to incorporate into the technology flow of, e.g., the electron gun fabrication, because of a several reasons: (1) low-throughput [83], (2) neither the synthesized CNT forest nor deposited CNTs are fully technologically compatible with the common thin-film technology to fabricate electronics [84], and (3) the emission is affected by the screening effect between vertically standing nanotubes [85].

A breakthrough to these issues was the development of an FE film cathode. It was made of dispersed CNTs in ethanol mixed into non-conducting epoxies [86]. Contrary to the vertical array, such CNT film/layer forms a mesh, i.e., randomly aligned CNTs, with a flattened surface with only a few jutting/protruding CNT tips (Figure 3a). The emission current was close to that of a single nanotube (0.1–10 μA vs. 0.1–1 μA) but at a higher bias voltage (200 V vs. 80 V) [80]. Research into this precursory film idea has been stopped for the next few years because of weak applicability, which is reflected in the lack of high brightness usually accomplished by a good electrical contact [87,88] and a high current [45]. One of the main reasons was the dependence of the emission current on the CNT density in the matrix [46]. The proof of this hypothesis was conducted by Nilsson et al. [85], who researched an FE cathode made by ink-jet printing CNT suspensions (Figure 3b,d).

On the other hand, in the early stages of research on FE CNT film cathodes, the low content of CNTs in the suspensions represented a significant obstacle. Typically, these suspensions contained CNT concentrations below 0.3 wt.% and were well-suited for spraying and ink-jet printing [89,90,91,92,93]. Due to the low density of these formed films, these techniques were not extensively used for FE cathode technology [94]. Nonetheless, research into film FE cathodes has continued to reveal attractive findings: (1) CNTs can emit from sidewalls (Figure 4) [95,96], and (2) despite CNT degradation during FE, the current level for a single CNT in a film is relatively high, from 300 nA to even 10 μA per emitter. This is enough to raise the substrate temperature toward the melting point (e.g., silicon [97]). The emission current depends on the type of CNTs and the methods used to synthesize them [48]. The higher crystallinity of the CNTs resulted in less Joule heating and led to improved stability and enhanced emission current density [98], as well as in brightness homogeneity [99].

This research had an impact on the development of a uniform and large area of FE electron sources formed in a simplified process like dip-coating, drop-casting, or filtration (Figure 5a,b) [98,100,101,102,103], and finally screen printing (Figure 5c) [104,105,106], which is an established and well-known method for fine patterning, with increasing use for flexible electronics [107,108,109], including vacuum electronics [110,111]. The graph summarizing the techniques for the FE cathodes upon the alignment of the CNTs and the scalability is presented in Figure 6 (author’s work).

### 4.1. Screen-Printed CNT-FE Cathodes

Screen printing is a well-established and significantly old technique for reproducibly writing patterns on flat surfaces [112]. The print industry has been using it to transfer black-and-white and color patterns on solid and non-solid materials [113], without any limits on the patterned area. The technique uses a wire-mesh frame that is placed over a fixed substrate. The wire-mesh design and material type [114] are two important factors in considering the affinity of the wire mesh, together with the choice of the substrate and suspension (paste, ink, or emulsion) [109,114]. The important parameter in screen printing is the viscosity of the suspension, which depends on the solvent: if it is too viscous, it will not transfer through the wire-mesh opening; if it is insufficient, it will not hold the pattern [114,115]. The fabrication of the suspension is also important. Commonly used ultra-sonication or harsh mechanical crushing is well-known to damage CNTs, diminishing the suspension cohesion [115]. In addition, damage to the CNTs has shown to have a negative influence on I–V characteristics of the FE cathodes (Figure 7) [105], as well as on the resistivity of the obtained film from the suspension and threshold field (adequate voltage) at which the emission starts [55].

The first report on screen CNT-FE cathodes came from Kwo et al. [104]. Initially, they synthesized multi-walled CNT (MWCNT) clusters by arc discharge between a pure graphite rod and a graphite disc in a helium atmosphere. They obtained micrometer-long CNT bundles and demonstrated FE condition with turn-on field (TOF) at 1.5 V/μm with an emission current density up to 7.3 mA/cm^2^. Next, the fibers were crushed in a ball mill for several hours, with the addition of binders (conductive pastes composed of carbonaceous particles), resulting in a CNT slurry. Finally, the slurry was screen-printed and heated (500 °C in air for 1 h), forming a CNT film. Next, its surface was etched in microwave Ar plasma for 30 s, in order to remove the binders. This improved emission characteristics (I–V). To demonstrate the potential large area patterning, the CNT cathode array was fabricated and worked as the FED (Figure 8a). The cathodes were characterized by cycling the I–V profiles, with similar emission current densities (1.5–6.5 mA/cm^2^) at 3 V/µm (Figure 8b) and lifetimes of about 1500 min (Figure 8b inset).

An interesting concept to combine screen printing and lithography patterning was presented by Bouchard et al. [116]. This was one of the earliest ideas on how to make a flat panel display greater than 30 inches (76 cm) in size, with the resolution limited to single micrometers. The CNT-FE cathode was fabricated in a three-electrode setup (triode), with various components. In a sequence of patterning steps similar to thin-film technology, all of the triode setup components were screen-printed, layer by layer, as follows: a conductive path, a dielectric separator, a gate electrode, and a layer of emitting material (a cathode). Additionally, all components were screen-printed, using suspensions, to which a photosensitive agent was added. In this way, a series of various photo imageable pastes were developed that differed by composition and weight percentage (wt.%) content. A similar process used for photolithography, e.g., in semiconductor technology, was used to pattern all triode components. The screen-printed films were UV irradiated through the mask to transfer the patterns and then baked. Next, the unnecessary material was removed, leaving the patterned area. If the films, made of various pastes, were screen-printed layer by layer, then all components could be patterned at once. The technology prevented the formation of electrical shorts between electrodes and assured that the components were finely aligned and adjusted vertically. The FE cathode arrays emitting light from the phosphors are presented in Figure 8c, with the current density up to about 1 × 10^−5^ A/cm^2^ (Figure 8d).

It has been found that the design of the field emitter determines its performance. Kwon and Lee [117] showed that the peripheral length of the patterned emitters cannot be neglected as emission primarily occurred from the edges. Hence, changes in the cathode design, from the high aspect ratio lines to the matrix of squares have resulted in more uniform and 1.4 times higher emission current, probably due to the non-uniform distribution of CNT tips from the surface of the film. Further experiments by following researchers [118,119] showed an increase of emission currents with the reduction of the emitter area and the spacing between them (Figure 9a,b). All the results indicated the potential of a microscale design and the importance of the fine patterning for CNT-FE cathodes.

Currently, photo imageable pastes can achieve tens of micrometer resolutions of the patterns and the separation distances if used in screen printing. However, UV exposure is still limited to thin films, as the UV irradiation cannot completely penetrate thicker films to reach cross-linking agents at the bottom. Moving forward, this issue in advanced processing was presented by Chung et al. [66], and the resulting CNT FED product is presented in Figure 1b. The cathode consisted of an array of field emitting CNT with a diameter of 20 μm on indium tin oxide (ITO) (Figure 10—an upper inset). Over the tips, there was an opening window with a diameter of 30 μm in a chromium film (Figure 10 a lower inset). Both the cathode and opening were electrically isolated by silicon dioxide (SiO_2_) and amorphous silicon (a-Si) films. The role of the chromium film with the opening was to extract electrons that diverged radially. First, the films were deposited and patterned for conducting pads (ITO), electrical isolation (a-Si, SiO_2_) and photo-masking (a-Si). After that, the chromium film with the electron extraction opening was made and covered by photo imageable CNT paste. Next, the paste was UV-exposed through the transparent back-side substrate, revealing the CNT cathode film. Nonetheless, the cathode thickness was only a few micrometers (4 μm), showing that it is indeed an example of a smart technique for photopatterning thick films surrounded by other components. This is particularly important, as the usage of CNT paste usually takes place at the end of the process flow to preserve its properties. The FED worked in a triode configuration: 1.5 kV and 100 V were applied to the anode and gate, respectively. The FED achieved over 0.4 mA (see Figure 10a).

Another, but less popular, technique that was reported as using the viscous CNT suspension is electro-plating [120]. In the process, a glass substrate was dipped in a nickel sulfate bath containing CNTs, leaving on the glass surface a film with field-emission properties. After dipping for 60 s at 80 °C, the film was dried and washed in deionized (DI) water. Finally, the sample was plasma-treated to remove the organic materials present in the suspension and the plating bath. The addition of CNTs made a film with a matrix of sharp edges and tips (Figure 10b inset), which is believed to contribute to the electric field enhancement factor. However, in this case, uniformity characterization of this coating proved to be difficult, as well as making patterns on the film, as in the previous examples. The FE measurements resulted in 1 mA/cm^2^ at 1.7 V/μm (Figure 10b). However, during the long emission (> 20 h), the characteristics became less stable, and the TOF almost doubled. The authors related this to the degradation of the film, i.e., heat-induced fracturing. Although the morphological changes increased the film resistivity and finally increased Joule heating, the cathodes operated for about 80 h at their current density, which is a significant result for potential applications.

### 4.2. CNT-FE Cathodes on the Tip of a Rod/Wire

Because X-ray imaging and therapy require a highly focused beam, the CNT-FE cathodes on the tip of a rod or wire were found to be the best structure for this. At the beginning of technology for an FE cathode on a tip, either electrophoresis or direct synthesis was used to deposit CNTs on the tip [121]. A single CNT was also mounted on a tip by a specialized piezo nano-manipulator [54]. The use of printable CNT pastes for an FE cathode on a tip was first presented by Kim et al. [2], who made a sub-millimeter-wide cathode film on the tip (dia.~ 800 µm) of a W rod (Figure 11a). The cathode of such a small dimension was made by using the contact method: The rod was simply covered by a viscous CNT paste that bonded physically to the tip. The authors wanted to cover only the cross-section of the tip surface in order to get a focused beam; hence, they used only a 1 uL droplet of a prepared suspension of CNTs mixed with Ag nanoparticles. Next, the dropped suspension was dried and thermally annealed. The annealing conditions (800 °C for 2 h under vacuum) melted the Ag nanoparticles that are believed to contribute to physical bonding between the W tip and CNTs (Figure 11b inset). This kind of FE cathode achieved 10 mA/cm^2^ at 1.15 V/µm (Figure 11b).

A similar strategy was approached by Sun et al. [123]. Following the other research results, they concluded that the addition of powder graphite to a CNT paste is a better choice to enhance adhesion to the surface tip. This is due to the similar nature of the materials bonded together, i.e., graphite powder and a graphite rod. The paste was dropped on a rod tip (diameter ~700 μm) and annealed in air at least 100 °C (The authors actually performed annealing at various temperatures and duration times.), to remove residual materials. After polishing the tip in order to protrude the CNTs, the cathode was ready for testing. The cathodes were made of pastes with three various ratios of CNTs to graphite powder: 10:100, 10:300, and 10:500 (Figure 11c). As expected, the highest content of graphite powder resulted in fewer CNTs being visible on the surface of the tip (Figure 11d inset). However, surprisingly, the best FE characteristic was obtained for the cathode with a moderate ratio of CNTs (10:300) (Figure 11d). This might be explained by the diminished screening effect for a looser density of CNTs, following the conclusion raised by Nilsson et al. [85], that there exists an intermediate regime defining the optimal inter-distance between CNTs. The highest emission current was 4.1 mA and the lowest TOF was 2.8 V/µm. Additionally, the performed lifetime tests exhibited over 20 h of working, with the current declining from 1.0 to 0.6 mA.

A smaller FE cathode on the tip of a Kovar wire was reported by Choi et al. [122], and it was made by using the tip contact method. Here, CNT suspension was brought in a precisely defined contact with the polished tip surface through Ohm measurement, involving a dedicated tool. By using this method, the authors were able to make a smaller FE cathode with a diameter of 50 μm (Figure 11e—upper view). After all necessary drying, removal of organic binders, and postprocessing, the authors presented the working cathode in the diode (Figure 11e—lower view) and triode (Figure 11f inset) configuration. In such a configuration, the cathode achieved an emission current of 220 μA (11.2 A/cm^2^) at 3.7 kV (Figure 11f).

Further research might be provided to improve the stability and to lower TOF and threshold field at high field-emission current compared to pristine CNTs, by analogy to the reported field emitters, where CNT were grown/sprayed on or between low-melting-point (below 500 °C) metals [124], metal oxides [125], and alloys [126].

Additionally, the local deformation of the substrate should be considered. Svensson et al. [127] noticed the substrate deformation (silicon dioxide) during field emission from a single-walled CNT (SWCNT). In their experiments, they intended to grow SWCNTs by the CVD with the presence of the low electric field (10–4 Vcm^−1^). After the growth process, they noticed the radial deformation around the location, where SWCNT was anchored. This deformation, which is a few nm in height and a few hundred nm in width, Svensson attributed to local melting of the SiO_2_ in a small region underneath the SWCNTs. Because at the tip of a single CNT, a field can be on the order of V/nm, it might induce additional local field emission from the tubes. This induces Joule heating with temperatures on the order of 2000 K. Together with electron bombardment, the SiO_2_ substrate lower its viscosity, which causes the liquid silica to flow locally. This additionally reveals the complex nature of the field emission.

## 5. Methods to Enhance Field Emission

### 5.1. Postprocess Treatments

The FE cathode made of a viscous suspension contains randomly distributed and aligned CNTs within a medium containing additives, such as binders (Figure 12a,b). This raises the problem of electrical contact between the CNTs and the substrate [128]. Hence, to realize the optimal performance of the cathode, a post-treatment process, such as to remove additive materials and improve material cohesiveness, is needed (Figure 12c,d). This increases the emission current level and, in most cases, reduces the emission TOF. In addition, the removal of residual organic binders and additive materials diminishes the amount of outgassing of the high surface area cathode material and reduces the probability of the formation of amorphous carbon within the cathode. These aspects lead to a decrease in the overall performance of the emitter. Hence, an important issue is to remove organic additives from the dried CNT film. Most of these evaporate at a temperature above 300 °C. If the annealing is done in the air, there is a fear of burning CNTs, which can normally withstand temperatures up to 750 °C [129,130]. Hence, researchers performed a thermal process in a high vacuum, as this helps to stick to the to the temperature threshold and makes the outgassing of residual material easier [122]. A strong electric field can also be used to orient the CNTs along the field directions permanently [131], but this has not been reported for films of randomly aligned CNTs.

Zhao et al. [133] and Shin et al. [105] used the idea that specific wavelengths (266 and 349 nm) can break the chemical bonds in organic binders. As a consequence, the emission sites in the temperature threshold make the outgassing of residual material easier [122]. CNT cathodes became activated (Figure 13a,b). The authors assigned the emission improvement to photodecomposition or photo-oxidation rather than to the photothermal effect. Indeed, as irradiation intensity increased, the emission current rose from 0.0027 to 14.45 mA/cm^2^, while the TOF decreased from 3.7 to 1.2 V/μm (Figure 13b). A laser was also used by Rinzler et al. [80] for oxidative etching nanotubes, to open them, which contributed to field-emission enhancement. A similar approach was used by Kim et al. [134]. They developed oxidative trimming where O2 reacted selectively with the highly emitting CNTs. According to the authors, the film consisting of CNTs with uniform height ensures the spatial uniform field emission (Figure 13c,d). During the experiments, they observed gradual etching of selected CNTs. This was reflected in I–V trend lines, which exhibited the current decrease and increase with the operational time (Figure 13d inset). Despite that, during the trimming, the emission current became 80 times smaller (from 4 mA down to 0.05 mA, after the third oxidative trimming cycle), the final cathode presented a remarkably uniform emission (Figure 13d) with a stable lifetime.

Vink et al. [132], following Dupont patent [116], presented a mechanical approach to improve I–V characteristics. Cathodes were screen-printed on Al-coated glass and Au-coated Si, using a commercial screen printer, then dried (at 120 °C) and annealed (at 400 °C) for 1 h in the air. Then the adhesive tape was applied to modify the film morphology; this simple process created a sparsely distributed array of vertically oriented CNTs, which was significantly sparser than what could be grown. Meanwhile, the binders detached from the cathode surface by tensile forces and adhered to the tape (Figure 12c,d). As a result, the cathodes achieved one thousand times higher current (0.5 vs. 500 mA/cm^2^ at 400 V) (Figure 14a,b), close to that which was achieved by the CNT forest [52,135] or a CVD-grown CNT array net [131] (10 mA/cm^2^), and two orders higher than the screen-printed film cathode post-treated with Ar plasma (7 mA/cm^2^) [104].

A similar postprocess treatment was also applied in [136,137,138,139]. However, it is difficult to discuss the effect of the treatment on emission due to a lack of data provided for comparison or other aspects influencing FE cathode properties, e.g., technology. Other methods were also investigated, such as the use of liquid elastomer [140], a soft rubber-roller [122,123,141,142,143], mechanical crush [144], and plasma [119,133,145,146,147]. The last one is believed not only to make CNTs protrude from the matrix (Figure 14c), but also to clear the cathode surface, improving the uniformity and reducing the cathode aging, and thus improving the FE characteristics (Figure 14d). In some instances, more than one technique was used [123,138,139,143]. For example, mechanical polishing and rubber rolling doubled the emission current (from 4.6 to 8.4 mA) [123]. On the other hand, the TOF increased from 2.8 to 3.2 V/µm, which the authors related to the shortened length of CNTs after the combined postprocess treatment methods.

From all reported postprocess treatment techniques, the most significant improvement was found by using poly-dimethylsiloxane elastomer (PDMS) [140]. Contrary to the other tools used in the mechanical approach, it offered several advantages: (1) It makes contact with a film surface with an inhomogeneous morphology, (2) it does not damage the CNTs, which ensures good electrical conductivity in the film, and (3) it can be applied for a complex structure containing a mask or a gate electrode, as was presented by the authors. The cathode was made from a mixture of CNTs, glass frits, and organic binders, screen-printed on an ITO glass, and dried in air (at room temperature for 10 min and at 150 °C for 1 h, and then at 300 °C.) and in a nitrogen atmosphere (at 400 °C for 30 min). Next, liquid PDMS was poured on the prepared cathode and cured at 150 °C for 10 min, to make it solid (Figure 15a). Finally, the solidified PDMS was detached from the cathode, leaving the CNT film surface with much microscale roughness (Figure 15b). The I–V characteristics improved, showing the rise of a nearly flat curve to a nonlinear shape with the emission current over 12 mA, with a uniform luminescence (Figure 15c). However, due to the lack of extensive data on optimizing the emission parameters, the possibility for significant improvement remains.

### 5.2. Fillers and Coatings

A filler might be understood as any type of material, other than the CNTs, that is not a solvent and has an insulating or conducting feature. The filler is added to a suspension during its preparation or postprocessing and forms solidified additives that can be geometrically defined. Common fillers are particles of nanometer or micrometer size (micro- or nano-particles), which benefit from a lower melting temperature (hundreds of °C) than their bulk form [148]. Fillers can be non-organic and organic, e.g., conductive pastes, glass frits, and metallic and polymer particles, as well as their composites and additives characterized by UV exposure sensitivity for fine patterning [105,149]. Fillers are added to the low content of CNTs in suspension, in order to serve as a bonding material for strengthening the film [150]. For example, fillers help to prevent field-dependent degradation, when the loosely bonded CNTs are extracted from the cathode by electrostatic force [97]. Fillers, then prevalent as binders, fill the empty spaces between the CNTs and the CNTs and the substrate. This improves the adhesion [2,123] necessary to perform mechanical post-treatment, ensures the recovery of the connection of CNTs which broke down due to Joule heating [72], but also enables us to avoid the arcing during emission [151]. It improves the field enhancement effects as a consequence of the protruding CNTs [139] or thermal stability, where fillers may play the role of oxidation-inhibiting compounds similar to the boron- or phosphorous-related compounds, as it has been presented by Floweri et al. [74], where the addition of Ni prevailed the degradation of the field emission.

The right choice of fillers and their ratio content have shown to prolong the lifetime of the cathode by as high as 10 times [143] (Figure 16a), which, in the case of carbonaceous particles, is up to 20 h [123] (Figure 16b), while for metallic particles added to the suspension, it is up to 100 h [143] (Figure 16a,c). The ratio balance can also improve the electrical conductivity of the film [139]. Thus far, film cathodes without fillers enable current densities of tens of μA/cm^2^, which results in up to 1 μA (the author of the reference paper calculated the effective current density based on works of [118] and assuming that the area included in calculation is an effective emission area.), and, in some cases, to 1 mA (as previous notes.). Such a level is sufficient for the purpose of a FED source. However, for some applications, such as micro-thrusters in spacecraft or microwave amplifiers, the required current density is 100 μA/cm^2^ or more [152]. Cui et al. demonstrated the successful use of metal nanoparticle fillers to the cathode material [153]. In this way, they achieved emission levels at about 33.9 mA, with an emission current density of 4.2 A/cm^2^ from a 0.8 mm^2^ area of the film cathode. Despite significant progress in the technology of FE cathodes since this report, this level of performance remains exceptional.

There are also unexpected trends in the research provided toward knowing how the fillers and their ratio influence the performance of CNT film cathodes. Shin et al. [139], by changing the ratio of binders (glass frit vs. Ag paste) showed that the CNT film cathode with the highest resistance resulted in the highest current density, following the highest field enhancement factor for this sample. The authors claimed that this was due to the processing, as Ag paste served as a catalyst for the oxidation of the CNTs during the heat treatment (at 390 °C in the air). Hence, for the sample with the higher weight ratio of Ag paste, more CNTs were damaged by oxidation, which was visible by the lower number of protruded CNTs over the cathode film surface. Sun [123] showed that a high content of fillers might result in fewer emission sites. On the other hand, a lower content of fillers decreased the emission current as a consequence of the screening effect caused by higher CNT density.

One of the recent approaches to improve the field-emission performance is coating the CNT film with a low work-function material, to form a composite cathode. The approach is based on earlier findings, whereas grown-in-CVD-process CNTs were coated with lower work-function metal nanoparticles, such as Cs [154], Ti [155], Ag [156], Al [157], In [158], or Ta [159], and metal oxides, e.g., titanium oxide (TiO_2_) [160], and resulted in lower turn-on electric field, and threshold electric field could be achieved. In the case of a dielectric addition to the CNT array, the threshold electric field might be lower about a few times. Due to the presence of high dielectric constant material, the screening effect between CNTs is reduced, together with the mechanical stress that was generated by Joule heating. However, the work function was reported to be about half of the CNT [98,161].

The relevant work to the FE films made of CNT suspension comes from Wu et al. [35] and Song et al. [162]. The first group formed screen-printed carbon nanotubes (CNTs) and coated it with TiO_2_. The Ni-F was chosen as a substrate (Figure 17a) because of good electrical and thermal conductivities, whereas three-dimensional (3D) structure possessing high porosity and specific surface area enhances mechanical adhesion between the CNTs film and substrates. Finally, it can be easily and commercially obtained. The TiO_2_ in a form of the gel was spin-coated on the CNT film and tested. The results showed a significant reduction of the turn-on electric field and threshold voltage after coating with TiO_2_: from 0.75 and 1.75 V/μm to 0.40 and 0.75 V/μm, respectively (Figure 17b). The cathode operated without noticeable degradation for about 5 h. Moreover, the substrate, because of its morphology, allows for the increasing of FE sites by spreading the CNTs and, in consequence, the current density.

The second group chose LiF/Al (ϕ ~ 3.0 eV) to deposit on the CNT film due to the material’s low work function, equal to about 3.0 eV, and its usefulness as an efficient electron extraction layer in organic light-emitting diodes and organic solar cells [162]. In addition to the previous work, the authors precisely defined the thickness of LiF to be constant and equal to 5 nm, while the Al film to be 1 or 3 nm thick (Figure 17c). Both films were thermally evaporated on the already screen-printed CNT film. In addition to the above conditions, there was also a sample without Al coating. Next, they evaluated their relation to the field-emission properties. The authors found that the increase of the Al film thickness from 1 to 3 nm actually decreased the field-emission properties and resulted in bare improvement, considering the lifetime tests. Further research here is needed to get more data to define the right trend and confirm what the authors of the referred paper claimed, that the field-emission results are correlated with the conductivity of the composite film.

## 6. Summary

This paper presents field-emission-electron sources that use carbon nanotubes as the electron-emitting material. Approaches to form the FE cathode of the electron source from CNT suspensions were described and compared, including synthesis, electrophoresis, and screen printing. From all of these techniques, so far only screen printing offers a simple and scalable approach to fabricate large area and uniform for emission cathodes or cathode arrays, including their different arrangements and shapes. Meanwhile, the development of modified techniques, e.g., a combination of screen printing and photolithography, the resolution of a patterned single line can achieve even tens of micrometers. It has been shown that a suspension was successfully used for a cathode on the tip of a wire or a rod to get a highly focused electron beam. Additionally, it has been found that CNTs could emit from their sidewalls too. In many cases, various postprocess treatment methods or the addition of fillers to the suspension level up the CNTs, which indeed improves field emission and the lifetime of the cathode compared to the as-made mesh cathode. Though the overall performance of the FE cathodes fabricated from CNT suspensions is slightly inferior to that of synthesized CNTs (see the following table), from this review and other referenced works, postprocess treatments and the addition of fillers are viable approaches to improve the emission characteristics of these cathodes. As Kim [134] reported, theoretically, only a 10% difference in height of CNT tips leads to an almost 90% difference in emission currents. Thus, the precise control of the emitter array is essential to provide spatial emission uniformity. It is possible to assess the uniformity by the postprocess treatment. Therefore, it is important to adjust the parameters of post-treatment, such as process duration. For example, etching might result in over-etching of CNTs.

Table 1 and Table 2 summarize the selected cathodes mentioned in this review. Table 1 includes FE cathodes in the form of a film made of the suspension grouped by the technology used to make the cathode. Table 2 includes the cathodes made of pristine CNTs grown directly on the substrate by CVD-based methods and grouped by the structure of the cathode. The tables include the materials used for cathodes, technologies, the performance of the cathodes, and adequate references. For the performance, the best values were chosen, particularly a subsequent threshold electric field, Eth, or voltage, Vth, required for the particular emission current or current density, and duration of emission following the lifetime test. As can be seen from the wide spectrum of referenced works, the screen printing is, so far, the efficient and alternative technique to the CVD and perhaps the best solution for the large-area fabrication of the FE arrays. Additionally, the increased interest in this technique pushes forward dynamic development in viscous suspensions containing novel materials.

## 7. Issues and Perspectives

### 7.1. Field-Emission Theoretical Model

Though in the literature, for film cathodes, there is a developed theoretical model which might explain to some degree the evolution and self-assembly of the system of CNTs during field emission [158], the readers should also note the ongoing scientific discourse on the right theoretical model that describes the field emission; the commonly used Fowler–Nordheim equations were found to be inadequate and need to be corrected. So far, it is proposed to use the corrected F–N equation named Murphy–Good equations [19] in order to prevent the research-integrity problem, as it was called by R. Forbes in his latest paper [25]. Additionally, the *Seppen–Katamuki* (*SK*) analysis might be used to obtain the exact work function of the emitter, as well as to extract geometrical parameters of the field emitter [42,43,44].

### 7.2. Screen Printing

The major obstacle of screen printing and viscous suspensions as material for technology is the outgassing caused by the paste components during operation under vacuum conditions. In addition, the rheology of the suspension to more properly characterize the suspension is also needed to identify the critical properties, e.g., viscosity, viscoplasticity, homogeneity, etc., that determine the structure of the desired printed pattern. Another important point is that not all CNTs are the same. For example, the electronic properties of CNTs have a strong relationship with their structure, and so far it is a challenge to grow CNTs with unique electronic properties on an out-of-laboratory scale [81], making it difficult to repeat their field-emission performance. Therefore, it is important to consider that cathode performance and their processability depend on the type of CNTs used [70], as well as on their crystallinity. It was reported that high crystallinity reduced Joule heating, improved emission stability, and enhanced emission current [94] and brightness homogeneity [99]. If the FE cathode is made of CNT suspensions, its performance depends not only on the used CNTs but also on the type of solvents (organic, resins, and acids) and additives (surfactant, organic, and non-organic nanoparticles), as well as the process condition and postprocess treatment needed either to remove the solvent and additives or to protrude or rearrange the CNTs.

Finally, homogenous dispersion of the high crystalline metallic CNTs with minimal damage for field-emission purposes is not a trivial task. This point ensures the CNT cathode film stability during emission [30].

### 7.3. Screening Effect and Side-Wall Emission

With recent data about sidewall electron emission, flat CNT cathodes might bring some advantages, although this needs to be confirmed. Contrary to field emission from a CNT forest, in field emission from a flat CNT cathode, electric field screening phenomena seems to not play a major role. However, this issue has been not yet been investigated for the CNT matrix.

### 7.4. Electron Beam Focusing

Experimental data have already shown that, in the case of X-ray tubes, the sharp and irregular shape of the cathode surface causes problems with focusing the beam. An improvement was established by using a flat cathode, consisting of a CNT mesh or matrix [10]. On the other hand, the same group showed how the side-attached CNTs work as an emitter, which has a bad effect on the focusing of the electron beam, again providing an argument for planar cathodes.

### 7.5. Film Adhesion and Stability

Another issue related to the film cathodes is their self-assembly during emission caused by electrodynamic force and experimentally presented for CVD-grown CNTs [167]. Moreover, it will be good to investigate the influence of the adhesive layer on the field-emission properties of the CNT film, as it is suggested by Lim et al. [124], especially because there is already evidence that the weak adhesion between CNT and substrate might lead to emission instability and lowering the overall emission performance [168,169].

### 7.6. Addition of Fillers and Coating

Experimental proof prevailed that the presence of the intercalated metallic particles might enhance and reduce the field-emission parameters, depending on their ratio and type of the particles used [139]. The interesting comparison is between the works with attempts: (1) to make CNT film cathode free of organic or dielectric particles, and, in case of aligned CNT arrays, (2) to percolate the CNT aligned array with a dielectric material, which might have a significant impact on field-emission enhancement, and coating the cathode with a metallic film possessing lower work function than CNT showed it might be the right trend to improve the field-emission properties of the CNT film cathodes [170]. Although it had an effect on the decrease of the threshold voltage, the drawback of this approach is even three orders lower current density compared to the film cathodes without metal covering (see, for example, Figure 17 vs. Figure 4, Figure 9 and Figure 13). The influence of the coating film thickness on field-emission properties, as well as its correlation to the composite conductivity, should be further investigated to confirm the line trends and its correlation with the composite conductivity. It might also contribute to a better understanding.

All of these listed challenges should be again considered in order to have a better understanding of the electron emission from film cathodes made of CNT suspensions.

## Figures and Tables

**Figure 1 micromachines-11-00260-f001:**
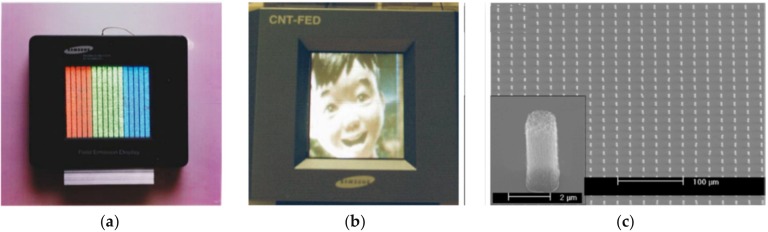
Examples of the field-emission displays (FED): (**a**) the emitting color (red–green–blue) phosphor columns image of fully sealed SWNT-FED presented by Samsung (reprinted with permission from [65]); (**b**) further results from Samsung presenting the color image (reprinted with permission from [66]); and (**c**) a scanning electron microscopy (SEM) image of CNT arrays for the FED by Philips (reprinted with permission from [67]).

**Figure 2 micromachines-11-00260-f002:**
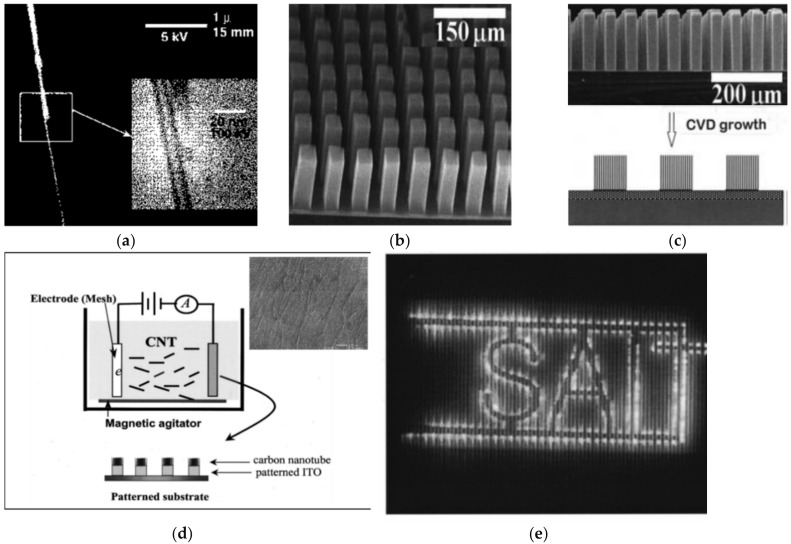
First approaches toward CNT field-emission (FE) cathodes: (**a**) a single CNT attached to a stalk—a high-resolution SEM image and transmission electron microscope (TEM) image (inset) (reprinted with permission from [80]); (**b**,**c**) synthesized and aligned arrays of CNT forest (reprinted with permission from [52])—SEM images of CNT towers (**b**) and its side-view variation (**c**), and the schematic cross-section illustration of the array (c—lower image); (**d**) CNT film deposited by electrophoresis—a schematic illustration presenting the process (reprinted with permission from [82]) with an SEM image of the deposited CNT by electrophoresis (insert view) (reprinted with permission from [81]); (**e**) the working FED from CNTs deposited by electrophoresis (reprinted with permission from [82]).

**Figure 3 micromachines-11-00260-f003:**
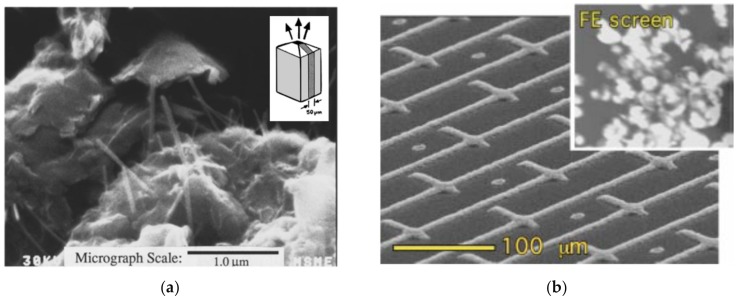
The FE cathodes made of a CNT matrix: (**a**) an SEM image of the dispersed CNT laminated onto a 50 × 50 μm^2^ area, the inset—the schematic illustration presenting the CNTs matrix laminated between two glass slides (reprinted with permission from [86]); (**b**) an SEM image of FE cathode made by ink-jetting of dispersed CNTs—the inset presents a macroscopic emission image of 2.5 × 2.5 mm^2^ on the phosphor screen; (**c**) the illustration presenting the field screening effect issue—a simulation of the electric field penetration depth for various CNT inter-distances; (**d**) SEM images (left) and FE maps (right) of the FE film cathodes, relevant to simulations presented in (**c**)—from the top to the bottom: the FE film with the highest, medium, and the lowest density of CNTs. Figure 3b–d reprinted with permission from [85].

**Figure 4 micromachines-11-00260-f004:**
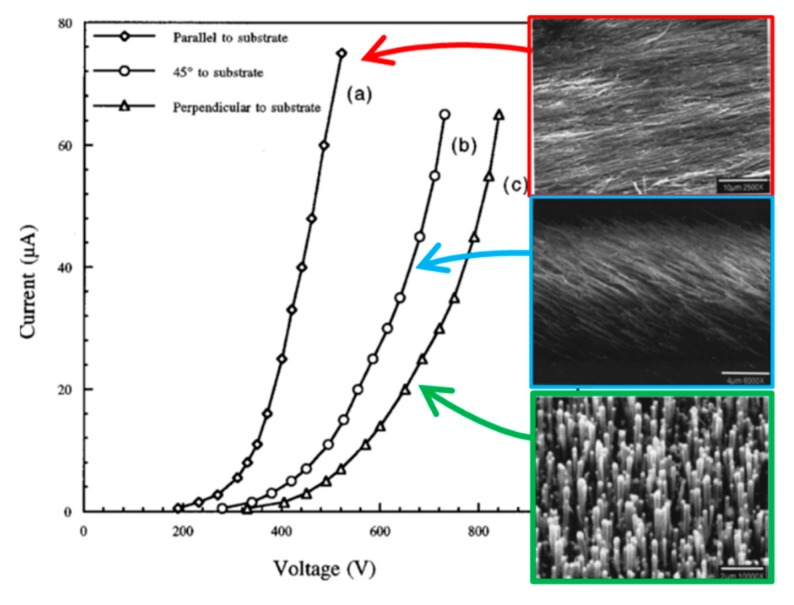
Field emission from the sidewalls of CNTs: the voltage–current (I–V) characteristic of films made of CNTs aligned parallel, at 45°, and perpendicular to the substrate (reprinted with permission from [95]).

**Figure 5 micromachines-11-00260-f005:**
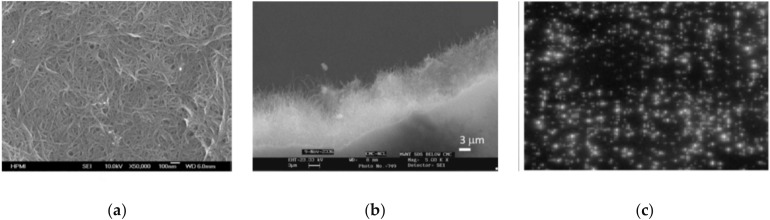
Uniform and large area FE cathodes: (**a**,**b**) SEM images of the buckypaper surface reprinted with permission from [102] (**a**) and reprinted with permission from [103] (**b**); (**c**) the emission image of the screen-printed FED (reprinted with permission from [105]).

**Figure 6 micromachines-11-00260-f006:**
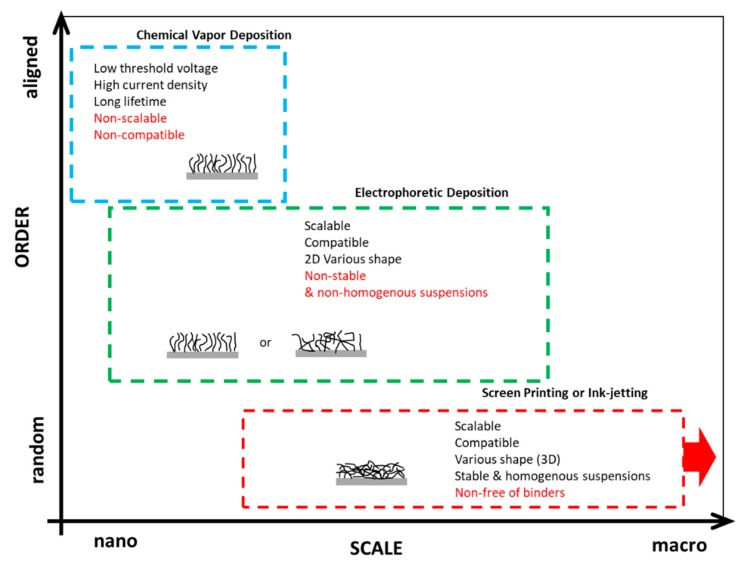
The graph presenting the possible scalability (horizontal axis) and the alignment of CNTs (vertical axis) obtained for various techniques to form FE cathodes (the author’s work).

**Figure 7 micromachines-11-00260-f007:**
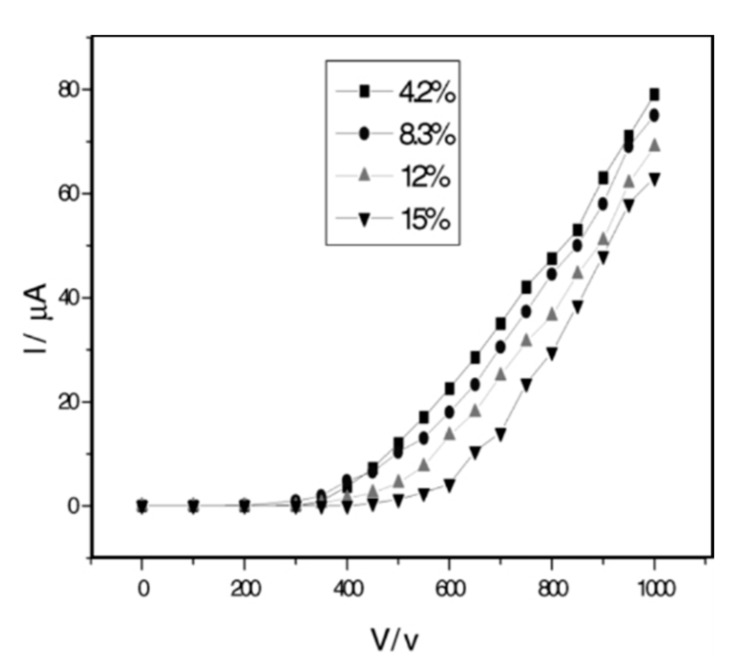
The graph showing the I–V characteristic of an FE cathode made of dried CNT suspensions for different constitutes of CNTs (reprinted with permission from [105]).

**Figure 8 micromachines-11-00260-f008:**
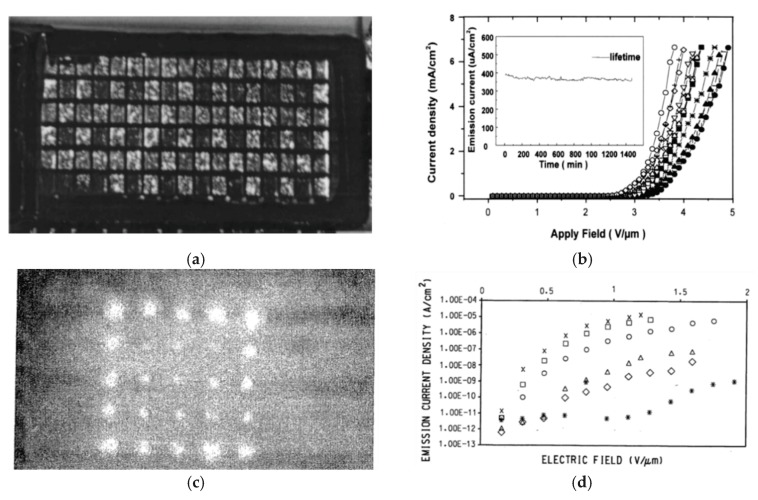
One of the first screen-printed CNT-FE arrays: (**a**,**c**) images of the light emitted from various phosphors by use of the screen-printed FE cathode arrays; (**b**,**d**) FE characteristics of several screen-printed CNT films; the inset in (**c**)—a lifetime test showing the stability of the emission current. (**a**,**b**) reprinted with permission from [104]; (**c**,**d**) reprinted from [116].

**Figure 9 micromachines-11-00260-f009:**
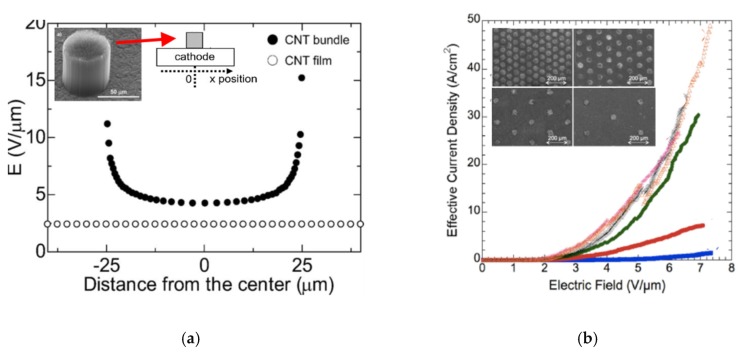
Influence of a design of a cathode on its emission: (**a**) the graph presenting simulation results for the FE cathode being a flat film and a bundle—the trend in electric field enhancement upon the distance from the center of the cathode, and the inset shows the bundle SEM image (left) and a schematic image (right); (**b**) the FE characteristic of the cathodes arrays with various distances between them, and the inset shows SEM images of these FE cathodes. Figure 9a,b reprinted with permission from [118,119], respectively.

**Figure 10 micromachines-11-00260-f010:**
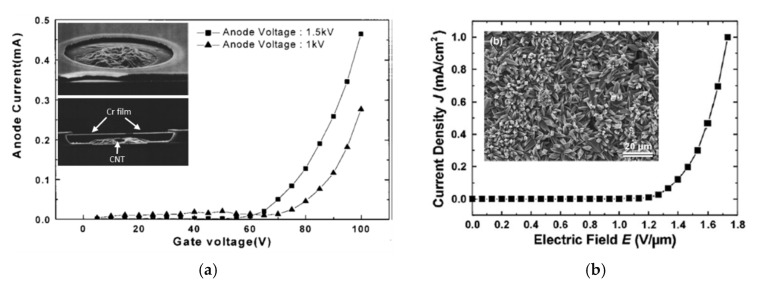
I–V characteristics of the CNT FED in a triode configuration. The insets show SEM images of a single FE cathode in a triode configuration. Figure 10a reprinted with permission from [66] and Figure 10b reprinted with permission from [120].

**Figure 11 micromachines-11-00260-f011:**
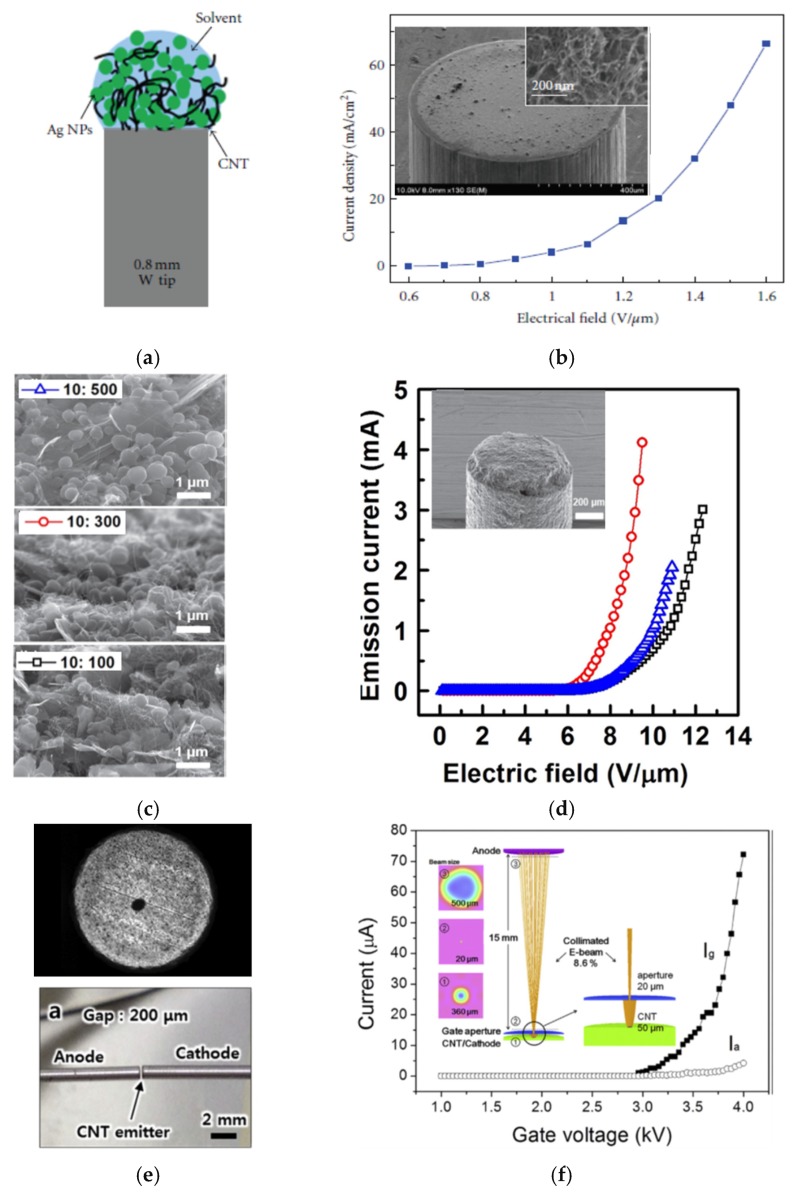
The CNT-FE cathode on tip: (**a**) a schematic illustration of the FE cathode on a tip, using a CNT suspension with silver nanoparticles (NPs); (**b**,**d**) the FE characteristic of the cathode, the inset—a zoom on a tip with drop-casted suspension; (**c**) SEM images of the surface of CNT cathode on a tip with various ratios of graphite powder and CNTs; (**e**) an OM image of the top of CNT-FE cathode (upper view); (**e**) the photography of the diode configuration, including the FE cathode on a tip (lower view); (**f**) I–V characteristic of the CNT FED in a triode configuration, and insets shows the simulated result of a collimated electron beam. Figure 11a,b from [2], Figure 11c,d from [116], and Figure 11e,f reprinted with permission from [122].

**Figure 12 micromachines-11-00260-f012:**
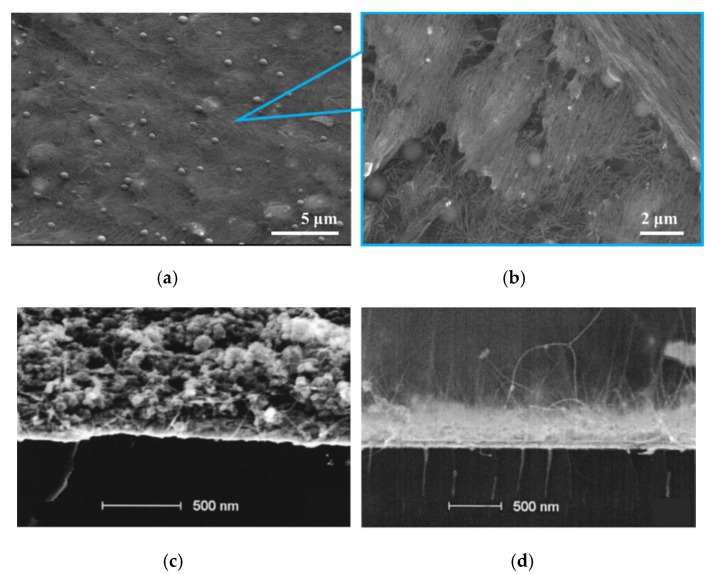
CNT matrix films made of CNT suspension: (**a**,**b**) The SEM images of the CNT film made of dried suspensions, between CNTs there are visible additives in a form of balls; (**c**) the SEM image of CNT film with additives and (**d**) after they were removed; Figure 12a,b from author’s work. Figure 12c,d reprinted with permission from [132].

**Figure 13 micromachines-11-00260-f013:**
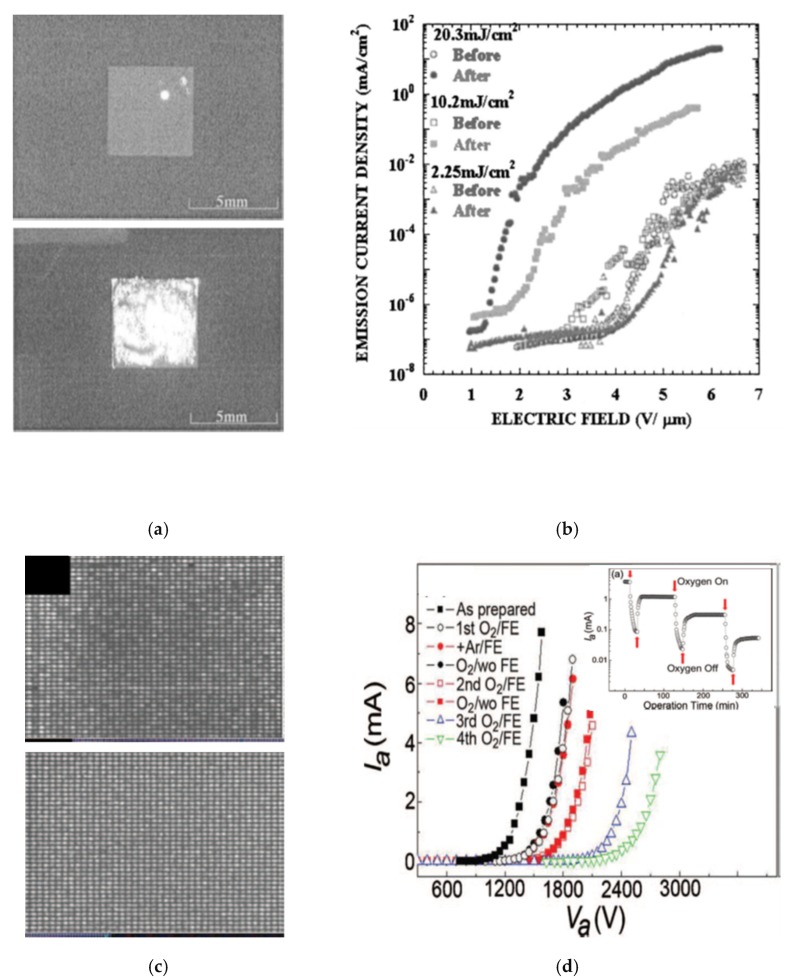
Emission enhancement by a postprocess treatment: (**a**,**c**) the pair of emission patterns from CNT FE cathode arrays before (upper view), and after laser irradiation (lower view); (**b**,**d**) the FE characteristic of the cathodes before and after laser irradiation. The inset in (**d**) shows the cathode operation at a periodic O2 supply, i.e., oxygen trimming is on or off. Figure 13a,b reprinted with permission from [133], and Figure 13c,d reprinted with permission from [134].

**Figure 14 micromachines-11-00260-f014:**
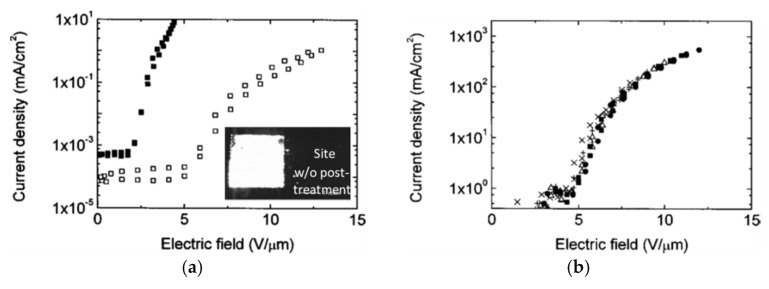
Emission enhancement by a postprocess treatment: the FE characteristics of a screen-printed CNT film before (**a**) and after the postprocess treatment (**b**). The inset in (**a**)—the emission site density image with (left) and without (right) postprocess treatment.Figure 14a,b reprinted with permission from [132].

**Figure 15 micromachines-11-00260-f015:**
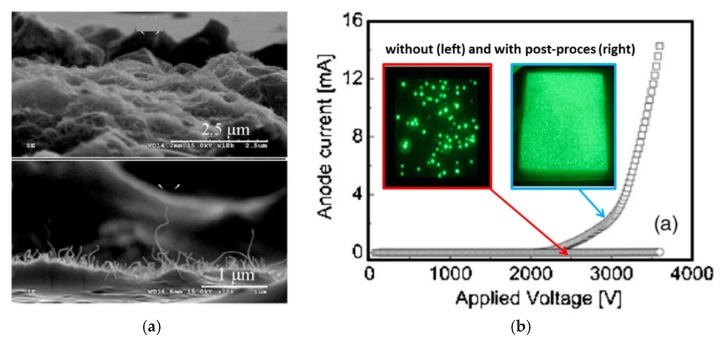
Emission enhancement by a postprocess treatment: (**a**) SEM images of the printed CNT films before (upper view) and after (lower view) postprocess treatment; (**b**) I–V characteristic of the cathode before and after postprocess treatment with the inset, showing emission images of the relevant cathodes reprinted with permission from [140].

**Figure 16 micromachines-11-00260-f016:**
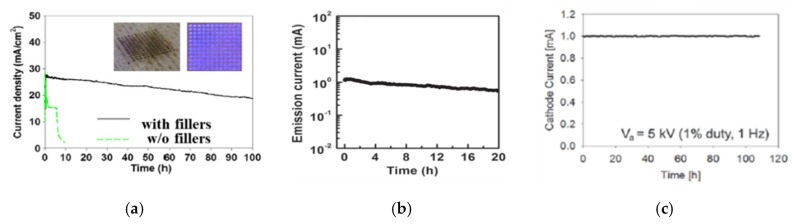
The lifetime tests, time vs. current, of the FE cathodes made of CNT pastes, with an addition of fillers. Figure 16a reprinted with permission from [143]. Figure 16b reprinted with permission from [123], and Figure 16c reprinted with permission from [142].

**Figure 17 micromachines-11-00260-f017:**
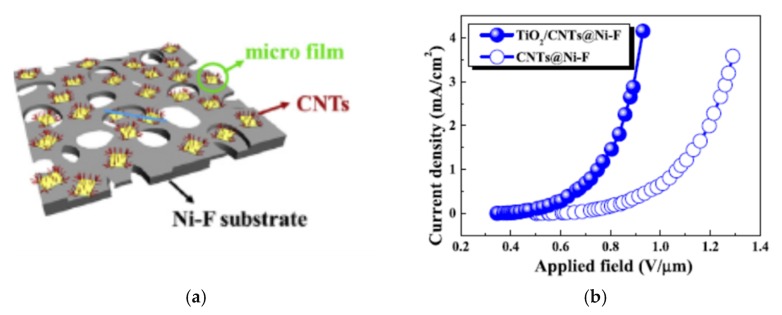
The field-emission enhancement by use of film coating with the material different from CNTs: (**a**) a schematic illustration of CNT FE cathode on Ni–Fe foam coated with TiO_2_ film and (**b**) its FE characteristics with and without TiO_2_ film. Reprinted with permission from [35].

**Table 1 micromachines-11-00260-t001:** The summary of the FE cathodes in a form of a film made of the suspension, grouped according the method to form a film.

CNT	Solvent, Additives/Fillers	Technology	E_th_ ^&^ (V/μm)	V_th_ ^%^ (V)	*t*^$^ (h)	*I* (mA/cm^2^)	*I* (μA)	Ref.
SW	Nitric and sulfuric acid/−	electrophoresis	−	900	18	−	10^3^	[82]
MW	unknown/−	Screen printing	1,5	~300	8000	10	~200	[23]
MW	polystyrene/−	Screen printing or casting	30	−	50	0,8	1	[55]
SW	IPA, nitrocellulose/-	Screen printing	3.0	−	−	90	1500	[65]
MW	EtOH, tributyl Phosphate, texanol, ethyl cellulose/Ni, TiO_2_	Screen printing	−	1500	10	−	5 × 10^3^	[74]
Un-known	unknown	Screen printing	3.0	−	25	0.4	−	[104]
MW	polyvinyl alcohol, dibutyl phthalate/frits	Screen printing	2.5	−	~1.5	35	−	[105]
Un-known	Unknown/unknown	Screen printing	−	2.5 (tri)	24	−	48.2	[106]
Un-known	Terpinol, organic binders, inorganic frits/−	Screen printing	1.5	−	100	1	−	[111]
DW ^#^	Organic binders/−	Screen printing	3.05	1220	−	−	20 × 10^3^	[128]
SW	Organic binders/−	Screen printing	~12.5	−	−	−	5 × 10^5^	[132]
MW	Texanol, acryl/Ni and TiO_2_ nanoparticles	Screen printing	2.2	−	−	−	1 × 10^5^	[138]
MW	Glass frits, organic binders/−	Screen printing	−	4500	−	−	10^3^	[139]
MW	Terpinol/Cu alloy and Al_2_O_3_ nanoparticles	Screen printing	−	3 × 10^4^ (tri.)	−	−	5 × 10^4^	[142]
DW	Ethyl cellulose/−	Screen printing	1.3	−	−	1	−	[146]
MW	Ethyl cellulose, terpineol/SiC and Ni nanoparticles	Screen printing	3500 (tri.)	−	−	20	10	[163]
Un-known	−/Bi and Na	Screen printing	6.5	−	−	4	3 × 10^4^	[152]
SW	Photo-sensitive vehicles/−	Screen printing and photo-lithography	−	80 (tri.) ^+^	−	−	~50	[66]
MW	Spin-on-glass, organic vehicle, photosensitive monomers, photosensitive oligomers, and photoinitiators	Screen printing and photo-lithography	2.46	−	−	9	−	[110]
Un-known	Ethyl cellulose, terpinol, photosensitive resin/Ag particles	Screen printing and photo-lithography	10	2000	−	0.2	−	[117]
Un-known	IPA, acrylate, cellulose, frit glass/SnO_2_	Screen printing and photo-lithography	3.6	−	12	2.0	−	[140]
MW	Texanol, photosensitive compounds, acryl/TiO_2_ microparticles and SnO_2_ nanoparticles	Screen printing and photo-lithography	17	−	−	50	−	[149]
MW	Organic binders/−	Screen printing and laser irradiation	6.2	−	−	20	−	[130]
MW	Photoimageable compounds/−	Screen printing and oxygen trimming	−	5000 (tri.)	120	−	200	[134]
SW	Organic binders/−	Screen printing and Ar plasma	−	1280	−	−	3 × 10^4^	[144]
MW	SiO_2_ sol, carboxymethyl cellulose, glycol/−	Screen printing and reactive ion etching	3.5	−	−	200	−	[145]
MW	1,2-dichloroethane/−	Spray coating	1.90	−	12	1.52	−	[94]
MW	1,2-dichloroethane/−	Spray coating	2.5	−	−	2	−	[124]
MW	EtOH	Deposition on a rod	4.6	−	~2	8.5	5.9 × 10^3^	[77]
SW	Sodium dodecyl sulfate	Forming the triangular shape from a filtered and dried paste	~1.5	−	20	100	22.4	[78]
MW	sodium dodecyl sulfate/graphite powder	Deposition on a rod	3.2	−	20	2 × 10^4^	8.5	[123]
SW	Water, surfactant	buckypaper	0.56	−	50	1	−	[102]
MW	Ferrocene-xylene	buckypaper	4.9	−	5	0.4	1	[103]
MW	nickel sulfate bath	electroplating	1.7	−	100	1	−	[117]
Un-known	Terpineol/Ni and SiC nanoparticles	Point contact method	−	3700	<1	1 × 10^4^	220	[122]
Un-known	EtOH, epoxy resin/−	A film was mounted between glass slides	−	200	−	400	0.1–10	[86]

^#^ MW = multi-walled nanotubes; in particular, DW = double-walled nanotubes; ^ SW = single-walled nanotubes; * CVD = chemical vapor deposition; ^&^ subsequent electric field required for the particular emission current or current density, beyond the emission initiation [45]; ^%^ subsequent voltage applied and required for the particular current or current density, beyond the emission initiation. (in case of the triode configuration, it is not a gate voltage); ^$^ it is the best-reported value in the reference; + tri. = triode configuration.

**Table 2 micromachines-11-00260-t002:** The summary of the remain FE cathodes, grouped according to the structure.

CNT	Structure	Technology	*E_th_* (V/μm)	*V_th_* (V)	*t* (h)	*I* (mA/cm^2^)	*I* (μA)	Ref.
Un-known	A single CNT	CVD	−	~700	−	−	2	[125]
MW	A single CNT mounted on W tip	−	−	319	−	−	1.1	[54]
Un-known	A single CNT with MgO coating	CVD and electron beam evaporation	−	1000	−	−	1	[125]
MW	Film	PECVD	4	−	−	−	10	[58]
Un-known	film	CVD	2.08	−	24	−	202	[121]
Un-known	film	CVD	6.5	−	125	10	−	[135]
Un-known	film	microwave PECVD	1.2	−	5	400	−	[126]
SW	Forest	CVD	1.85	−	12	1.25	−	[94]
MW	Bundle forest	PECVD	4.05 *	−	7	7	~1	[164]
Un-known	Bundle forest	PECVD	−	−	−	−	630	[165]
SW	Forest parallel aligned to the substrate	HFCVD	−	520	−	−	~71	[95]
SW	Bundle forest	Arc discharging	−	1200	−	−	0.1	[23]
MW	Radially aligned tubes in arrays	Spray pyrolysis	0.78	−	>14	7.71	−	[31]
MW	Forest pitch array	CVD *	4.8–6.1	−	20	10	−	[52]
MW	CNT array in a shape of a star	CVD	−	1000	368	9.08	−	[63]
Un-known	CNT array	CVD	5.33	−	−	50	2 × 10^4^	[67]
MW	Uniform array of individual tubes	PECVD	−	−	−	−	~20	[87]
Un-known	Patterned forest as a bundle	CVD and photolithography	2.4	−	200	150	3	[118]
MW	Patterned forest arrays	CVD	6.7	−	3300	−	5000	[119]
MW	Patterned forest arrays	thermal CVD	−	600	10	10^−2^	2230	[166]
SW	CNT wire mesh	CVD	1.5	−	−	−	10^3^	[136]
MW	Yarn	-	0.15–0.5	500–1100	−	−	~600	[56]
Un-known	Fiber	Wet spinning	−	850	−	−	3.5	[88]
MW	fiber	Twisted CNT yarns	−	750	−	−	1500	[76]

* The authors of [166] determined V_th_ at 1 mA/cm^2^.

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
