# Peer review of "Field Emission Cathodes to Form an Electron Beam Prepared from Carbon Nanotube Suspensions"

_micromachines, 2020, doi:10.3390/mi11030260_

Round 1
Reviewer 1 Report
In this paper, the authors report a FE cathode. FE cathodes with CNT are reviewed by various explanation. Very good results of CNT cathodes comparison such as FE performance survey are reported.
1. From the lines 97 to 108, the author describes about the field emission set-up. However, the author fails to mention the rough value (lower than 10-7 torr) of vacuum pressure needed for stable emission of electron emission inside the chamber or device.
In the line 100, it is mentioned that (The triode configuration forms an electron gun). I think it should be written as (the triode and diode configuration both can be regarded as an electron gun.)
2. In figure 3 (C)., The author mentions about the screening effect between emitters. However, the top (low-density CNTs) and bottom (high-density CNTs) pictures seem to be swiped. The adjacent SEM images next to the simulation results do not match.
3. It would be better if author provide citations to the lines 137~139 (Especially, as in recent reported scientific discussion, it was proposed to use the modified F-N equations named Murphy-Good equations and the Schottky-Nordheim barrier that represent better physics to explain and interpret the FE characteristics.)
4. Please revise the text according to attached file.
5. Please refer the below references in Table 2.
[1] J. Lim, A. P. Gupta, S. Yeo, M. Mativenga, M. Kong, C.-G. Cho, J. Ahn, S. H. Kim, and J. Ryu, “Design and Fabrication of CNT-Based E-Gun Using Stripe-Patterned Alloy Substrate for X-Ray Applications,” IEEE Trans. Electron Devices, vol. PP, no. 99, pp. 1–4, Oct. 2019.
[2] S. Park, A. P. Gupta, S. Yeo, J. Jung, S. H. Paik, M. Mativenga, S. H. Kim, J. H. Shin, J. Ahn, and J. Ryu, “Carbon Nanotube Field Emitters Synthesized on Metal Alloy Substrate by PECVD for Customized Compact Field Emission Devices to Be Used in X-Ray Source Applications,” Nanomaterials, vol. 8, no. 6, May 2018.
[3] A. P. Gupta, S. Park, S. J. Yeo, J. Jung, C. Cho, S. H. Paik, H. Park, Y. C. Cho, S. H. Kim, J. H. Shin, J. S. Ahn, and J. Ryu, “Direct Synthesis of Carbon Nanotube Field Emitters on Metal Substrate for Open-Type X-ray Source in Medical Imaging,” Materials, vol. 10, no. 8, p. 878, Jul. 2017.

Author Response
The author would like to thank the reviewer for valuable comments. Please find the response below.
Point 1: 1. From the lines 97 to 108, the author describes about the field emission set-up. However, the author fails to mention the rough value (lower than 10-7 torr) of vacuum pressure needed for stable emission of electron emission inside the chamber or device.
In the line 100, it is mentioned that (The triode configuration forms an electron gun). I think it should be written as (the triode and diode configuration both can be regarded as an electron gun.)
Response 1: The author included the Reviewer’s notice as follows:
“For the basic evaluation, the FE electron sources are measured in a diode configuration, i.e. cathode-anode with a gap between them with vacuum pressure below 10-7 Torr in a device or chamber for stable electron emission. Such configuration is faster and cost-efficient compared to the triode configuration, which is formed by a cathode, an extraction electrode/gate, and an anode. The triode and diode configuration, both can be regarded as an electron gun [27].”
Point 2. In figure 3 (C)., The author mentions about the screening effect between emitters. However, the top (low-density CNTs) and bottom (high-density CNTs) pictures seem to be swiped. The adjacent SEM images next to the simulation results do not match.
Response 2: The author corrected Figure 3.
Point 3. It would be better if author provide citations to the lines 137~139 (Especially, as in recent reported scientific discussion, it was proposed to use the modified F-N equations named Murphy-Good equations and the Schottky-Nordheim barrier that represent better physics to explain and interpret the FE characteristics.)
Response 3: The author included the reference.
Point 4. Please revise the text according to attached file.
Response 4: The author noticed the suggestions of the reviewer and made the corrections referring to points 1 and 2 as well as yellowish labeled text lines.
Point 5. Please refer the below references in Table 2.
[1] J. Lim, A. P. Gupta, S. Yeo, M. Mativenga, M. Kong, C.-G. Cho, J. Ahn, S. H. Kim, and J. Ryu, “Design and Fabrication of CNT-Based E-Gun Using Stripe-Patterned Alloy Substrate for X-Ray Applications,” IEEE Trans. Electron Devices, vol. PP, no. 99, pp. 1–4, Oct. 2019.
[2] S. Park, A. P. Gupta, S. Yeo, J. Jung, S. H. Paik, M. Mativenga, S. H. Kim, J. H. Shin, J. Ahn, and J. Ryu, “Carbon Nanotube Field Emitters Synthesized on Metal Alloy Substrate by PECVD for Customized Compact Field Emission Devices to Be Used in X-Ray Source Applications,” Nanomaterials, vol. 8, no. 6, May 2018.
[3] A. P. Gupta, S. Park, S. J. Yeo, J. Jung, C. Cho, S. H. Paik, H. Park, Y. C. Cho, S. H. Kim, J. H. Shin, J. S. Ahn, and J. Ryu, “Direct Synthesis of Carbon Nanotube Field Emitters on Metal Substrate for Open-Type X-ray Source in Medical Imaging,” Materials, vol. 10, no. 8, p. 878, Jul. 2017.
Response 5: The author included listed above references in Table 2.
Reviewer 2 Report
This manuscript describes the recent results according to electron emission source architecture to progress electrical and artificial properties of carbon nanotubes (CNTs) as field emitters, and evaluates field emission characteristics with them. The paper shows some artificial methods for electrical source, especially for wet coating processes and their synthesized morphology of CNTs, and field emission theoretical model and protocol.
<Comment>
In the chapter of field emission theoretical model, the corrected F-N equations was proposed with thermionic field emission and the transition region between cold and thermionic cathode by E. L. Murphy et al., in the manuscript. The late theoretical papers based on thermionic field emission have already been reported by Y. Gotoh et al. If possible, the reviewer recommends for their reports (an example; ref. R1) to consider in your manuscript.
Furthermore, in field emission, the reviewer thinks that it is necessary for the preparation of an artificial field emitter device employing CNTs to improve the emission homogeneity with a high current output, long durability, and high-energy consumption efficiency in our habitat scene. Is it important for artificial field emitters to control the crystallinity of electron emission material as field emitter? Some references (an example; ref. R2) mentioned the control of a crystallization of CNTs should be important to construct an electrical device for practical realization.
References;
R1. Y. Gotoh, W. Ohue, and H. Tsuji, “Characterization of the electron emission properties of hafnium nitride field emitter arrays at elevated temperatures.” J. Appl. Phys. 121, 234503 (2017).
R2. N. Shimoi, A.L. Estrada, Y. Tanaka, and K. Tohji, “Properties of a field emission lighting plane employing highly crystalline single-walled carbon nanotubes fabricated by simple processes.” Carbon 68, 228–235 (2013).
For above reason, this paper needs revisions for the publication standard of this journal at the moment.
Author Response
The author would like to thank the reviewer for the valuable comments and pointing significant issues concerning the crystallynity influence on field emission and Seppen-Katamuki analysis. Please find the extended response below.
Point 1: In the chapter of field emission theoretical model, the corrected F-N equations was proposed with thermionic field emission and the transition region between cold and thermionic cathode by E. L. Murphy et al., in the manuscript. The late theoretical papers based on thermionic field emission have already been reported by Y. Gotoh et al. If possible, the reviewer recommends for their reports (an example; ref. R1) to consider in your manuscript
Response 1: The author included the Reviewer’s notice as follows, starting in the line 140:
“ It should be also recalled here Seppen-Katamuki (SK) analysis, that enables to obtain the exact work function of the emitter as well as to extract geometrical parameters of the field emitter [41]; for example, SK analysis has been used to evaluate the changes in work function at elevated temperature [42] or to derivate the length of carbon nanotubes in the field emission arrays [43]. “
With the following references:
[41] Gotoh, Y.; Nozaki, D.; Tsuji, H.; Ishikawa, J.; Nakatani, T.; Sakashita, T.; Betsui, K. Significant improvement of the emission property of Spindt-type platinum field emitters by operation in carbon monoxide ambient. Appl. Phys. Lett., 2000, 77(4), 588-590.
[42] Gotoh, Y.; Ohue, W.; Tsuji, H. Characterization of the electron emission properties of hafnium nitride field emitter arrays at elevated temperatures. J. Appl. Phys., 2017, 121, 1-12.
[43] Gotoh, Y.; Kawamura, Y.; Niiya, T.; Ishibashi, T.; Nicolaescu, D.; Tsuji, H.; Ishikawa, J.; Hosono, A.; Nakata, S.; Okuda, S. Derivation of length of carbon nanotube responsible for electron emission from field emission characteristics. J. Appl. Phys., 2007, 90, 1-3.
And starting in the line 651 (summary):
“Additionally, Seppen-Katamuki (SK) analysis might be used to obtain the exact work function of the emitter as well as to extract geometrical parameters of the field emitter [41-43].”
Point 2. Furthermore, in field emission, the reviewer thinks that it is necessary for the preparation of an artificial field emitter device employing CNTs to improve the emission homogeneity with a high current output, long durability, and high-energy consumption efficiency in our habitat scene. Is it important for artificial field emitters to control the crystallinity of electron emission material as field emitter? Some references (an example; ref. R2) mentioned the control of a crystallization of CNTs should be important to construct an electrical device for practical realization.
Response 2: The author exended the comment concerning the influence of crystallinity of the CNTs on the emission, starting in the line 252:
“The higher crystallinity of the CNTs resulted in less Joule heating and led to improved stability and enhanced emission current density [97] as well as in brightness homogeneity [98].”
With the reference added - [98]:
[98] Shimoi, N.; Estrada, A. L.; Tanaka, Y.; Tohji, K. Properties of a field emission lighting plane employing highly crystalline single-walled carbon nanotubes fabricated by simple processes. Carbon, 2013, 65, 228-235.
And the author extended the cooment in the line 661 (summary):
“Therefore, it is important to consider that cathode performance and their processability depends on the type of CNTs used [69], also on their crystallinity. It was reported that high crystallinity reduced Joule heating, improved emission stability and enhanced emission current [97] and brightness homogeneity [98]. ”